# Cryo-EM structure of coronavirus-HKU1 haemagglutinin esterase reveals architectural changes arising from prolonged circulation in humans

Daniel L. Hurdiss [1,2✉], Ieva Drulyte[3,5], Yifei Lang[1,5], Tatiana M. Shamorkina[4], Matti F. Pronker [4], Frank J. M. van Kuppeveld [1], Joost Snijder[4] & Raoul J. de Groot[1✉]

The human betacoronaviruses HKU1 and OC43 (subgenus Embecovirus) arose from separate zoonotic introductions, OC43 relatively recently and HKU1 apparently much longer ago. Embecovirus particles contain two surface projections called spike (S) and haemagglutinin-esterase (HE), with S mediating receptor binding and membrane fusion, and HE acting as a receptor-destroying enzyme. Together, they promote dynamic virion attachment to glycan-based receptors, specifically 9-O-acetylated sialic acid. Here we present the cryo-EM structure of the ~80 kDa, heavily glycosylated HKU1 HE at 3.4 Å resolution. Comparison with existing HE structures reveals a drastically truncated lectin domain, incompatible with sialic acid binding, but with the structure and function of the esterase domain left intact. Cryo-EM and mass spectrometry analysis reveals a putative glycan shield on the now redundant lectin domain. The findings further our insight into the evolution and host adaptation of human embecoviruses, and demonstrate the utility of cryo-EM for studying small, heavily glycosylated proteins.

[1] Virology Section, Infectious Diseases and Immunology Division, Department of Biomolecular Health Sciences, Faculty of Veterinary Medicine, Utrecht University, Yalelaan 1, 3584 CH Utrecht, The Netherlands. [2] Cryo-Electron Microscopy, Bijvoet Center for Biomolecular Research, Department of Chemistry, Faculty of Science, Utrecht University, Padualaan 8, 3584 CH Utrecht, The Netherlands. [3] Materials and Structural Analysis, Thermo Fisher Scientific, Achtseweg Noord 5, Eindhoven 5651 GG, The Netherlands. [4] Biomolecular Mass Spectrometry and Proteomics, Bijvoet Center for Biomolecular Research, Department of Chemistry, Faculty of Science, Utrecht University, Padualaan 8, 3584 CH Utrecht, The Netherlands. [5]These authors contributed equally: Ieva Drulyte, Yifei Lang. ✉email: d.l.hurdiss@uu.nl; r.j.degroot@uu.nl

Coronaviruses (CoVs) are enveloped positive–sense single–stranded RNA viruses of mammals and birds with a propensity to cross host species barriers[1]. Zoonotic CoV infections pose an ever-looming threat to public health[2]. Indeed, this century alone saw the advent of three novel respiratory CoVs highly lethal to humans. The 2002/2003 SARS-CoV variant was quickly contained[3,4], and MERS-CoV, natural to dromedary camels, does not spread efficiently within the human population[5]. SARS-CoV-2, however, seems well on course to become established in humans[6]. Four other respiratory coronaviruses of zoonotic origin did breach the species barrier successfully and are maintained worldwide through continuous human-to-human transmission[7]. Studies of these latter human CoVs and their zoonotic ancestors are instrumental in understanding the odds and risks of CoV cross-species transmission as well as the requirements for adaptation to the human host.

Human CoVs HKU1 and OC43 (subgenus *Embecovirus*, genus *Betacoronavirus*) are related but distinct[8]. In immunocompetent individuals, these viruses are generally associated with common colds, but may cause significant morbidity and even mortality in the frail[9,10]. They entered the human population separately: HKU1 presumably several hundred years ago from a yet unknown animal reservoir[11,12], whereas OC43 entered far more recently (70–120 years ago), apparently from a bovine coronavirus (BCoV) spill-over[13]. These embecoviruses use 9-*O*-acetylated sialoglycans (9-*O*-Ac-Sia) as receptors to which they attach in a highly dynamic fashion with fast on and off rates via fusion spike protein S[14,15]. A second type of envelope protein unique to embecoviruses, called the haemagglutinin-esterase (HE), serves as a receptor-destroying enzyme[8,16].

CoV HEs are 40–50 kDa type I membrane glycoproteins that assemble into covalently-linked homodimers, disulfide–bonded in the juxta-membrane region[17]. In HEs of animal embecoviruses, an *O*-Ac-Sia-binding lectin domain (LD) appended to the esterase domain (ED) upregulates sialate-*O*-acetylesterase activity towards multivalent ligands such as the densely clustered sialoglycans on mucins[8,17–19]. However, OC43 and HKU1, subject to convergent evolution, lost HE lectin function[8]. Thus, the dynamics of virion-glycan interactions were altered, and virion-mediated receptor destruction was restricted, apparently as an adaptation to replication in the human respiratory tract. For OC43, the phylogenetic record shows that after its introduction in humans (Supplementary Fig. 1), HE-mediated receptor binding was selected against and ultimately lost completely through progressive accumulation of subtle single site substitutions in LD. In HKU1, HE lectin function was also lost, but here the LD underwent several deletions[8,20].

For OC43, BCoV and murine embecoviruses[8,17–19], HE crystal apo- and holo-structures revealed how HE LDs bind *O*-Ac-Sias and how this property was lost in OC43. HKU1 HE, however, proved refractory to crystallisation. Exhaustive attempts to solve its structure and assess the structural consequences of the LD deletions did not meet with success, prompting us to consider alternative methods. In the last decade, single-particle analysis by cryo-electron microscopy (SPA cryo-EM) revolutionised the field of structural biology with methodological advances now allowing the visualisation of large biological macromolecules at resolutions which permit atomic model building[21]. In the wake of these innovations, cryo-EM structures were determined for several CoV S proteins, including those of OC43 and HKU1 to resolutions of 2.8 and 4 Å resolution, respectively[15,22]. These studies were greatly facilitated by the considerable size (>400 kDa) and three-fold symmetry of the S homo-trimers. Because of the low signal-to-noise ratio of cryo-EM images, SPA reconstruction of specimens smaller than 100 kDa remains problematic and only for a handful of such proteins, high-resolution (i.e. <3.5 Å) cryo-EM

structures have been resolved[23–25]. Nevertheless, inspired by these reports, we explored SPA to characterise the homo-dimeric HKU1 HE ectodomain, a complex with an ordered protein mass of 76 kDa plus ~30 kDa of *N*-glycans, of which 11 kDa are ordered.

Here we present the structure of HKU1 HE at a global resolution of 3.4 Å. The data reveal that, over centuries of HKU1 circulation in humans, the variable loops that form the LD receptor-binding site were trimmed back completely. The jelly roll core structure, on which these loops were grafted, remains intact, presumably to preserve structure-function of the ED SGNH hydrolase into which it is embedded. Conceivably, the conservation of a functionally obsolete but structurally essential domain in a viral envelope protein may pose an antigenic liability. The data show that through acquisition of new *N*-linked glycosylation sites, the HKU1 HE LD domain was provided with a putative glycan shield. The findings provide structural insight into the evolution of multidomain proteins after partial loss of function and a predictive evolutionary trajectory for OC43 HE.

## Results

**Structure determination of HKU1 HE by cryo-EM.** There are two variants of the HKU1 HE protein, type A and type B, which share ~85% sequence identity[26]. For structural analysis, the HKU1-A HE ectodomain was expressed as an Fc fusion protein in HEK293T cells and purified by protein A affinity chromatography, followed by on-the-bead thrombin cleavage to remove the Fc domain, and size exclusion chromatography (Supplementary Fig. 2A and Supplementary Table 2). The protein backbone of the HKU1 HE ectodomain construct has a molecular weight of ~40 kDa but runs as a smeared band between 50 and 75 kDa by SDS-PAGE analysis (Supplementary Fig. 2B), indicating extensive and heterogeneous glycosylation. Optimised cryo-EM grids, prepared with purified HKU1 HE, revealed monodisperse particles in a variety of orientations (Fig. 1a). Single-particle analysis produced 2D class averages corresponding to different views of the HKU1 HE dimer (Fig. 1b). Subsequently, a 3D reconstruction of HKU1 HE was produced at a global resolution of 3.4 Å (Fig. 1c and Supplementary Figs. 3 and 4A, B). Local resolution of the map ranges from 3.3–4.1 Å with the most well-resolved regions located in the core of the molecule (Supplementary Fig. 4D). The quality of the map permitted modelling of residues 15–346 (Fig. 1d, e). Each protomer is stabilised by six disulfide bonds, the presence of which are corroborated by the cryo-EM density (Fig. 1f). Dimerisation of HKU1 HE is achieved by interaction of the vestigial LDs and the membrane-proximal domains (MP) (Fig. 2a, b).

**Comparison of HKU1 HE to those of related embecoviruses.** The overall fold of HKU1 HE is like those of related murine coronaviruses (MCoVs) and BCoV, which have functional lectin domains (Fig. 2b–d)[17,19]. Indeed, comparison of the membrane-proximal domain and esterase domain of HKU1 to these related proteins revealed a high-level of structural conservation, with the aligned Cα positions having RMSD values of 0.81 Å and 0.76 Å for MCoV-New-Jersey strain (MCoV-NJ), and 0.83 Å and 0.84 Å for BCoV-Mebus strain. While the core fold of the esterase domain is similar between all three proteins, the α2 helix of HKU1 more closely resembles that of BCoV[17], and lacks the zinc-binding site found in MCoV-NJ[19] (Fig. 2b–d). The ED catalytic site of HKU1 HE is most similar to that of BCoV, OC43 and MCoV-DVIM strain (Supplementary Fig. 5A, B), which all recognise 9-*O*-Ac-Sia. This is consistent with functional data showing that HKU1 HE possesses sialate-9-*O*-acetylesterase receptor-destroying activity[8]. In contrast, MCoV-NJ HE recognises 4-*O*-Ac-Sia, thus

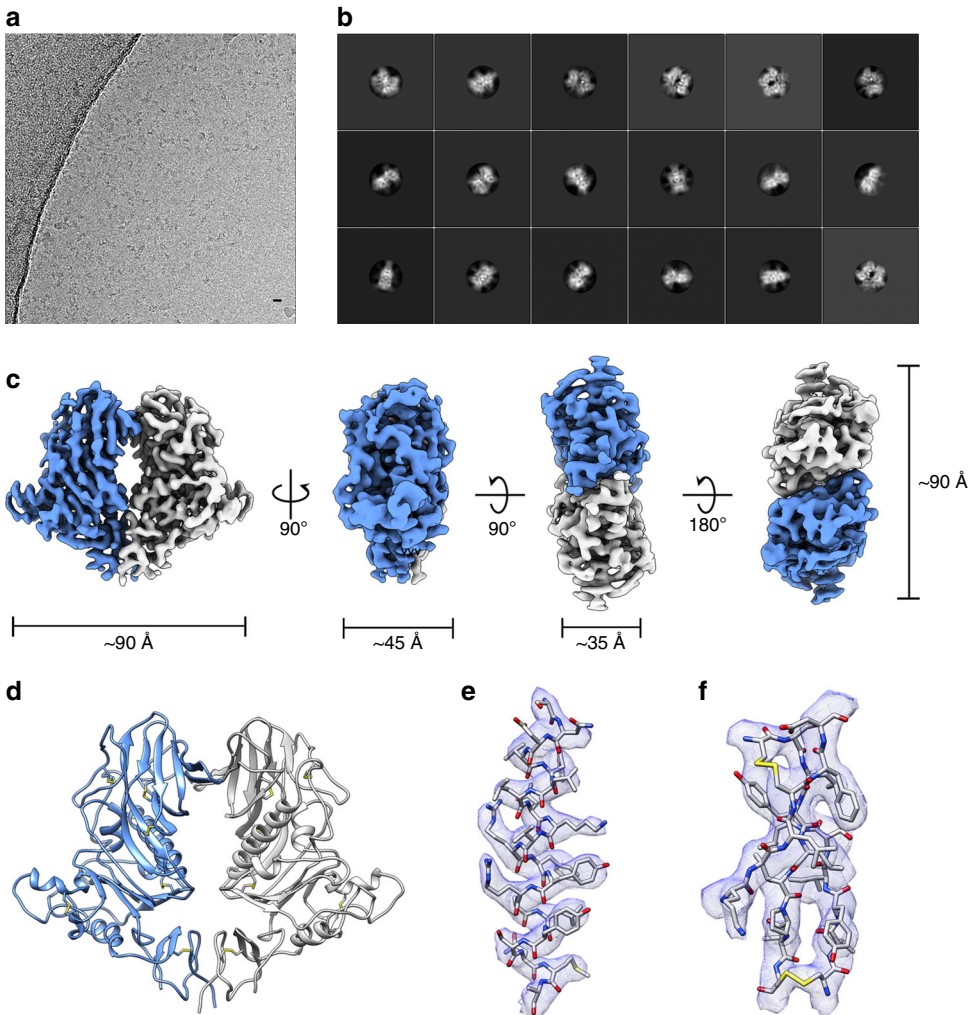

**Fig. 1 Structure determination of HKU1 HE by single-particle analysis. a** Representative motion-corrected electron micrograph of HKU1 HE embedded in vitreous ice. Scale bar = 10 nm. **b** Representative reference-free 2D class averages. **c** Orthogonal views of the HKU1 HE EM density (coloured by subunit). **d** Cartoon representation of the atomic model of the dimeric HKU1 HE complex. **e** EM density (blue mesh) zoned 2 Å around an α-helix comprising residues 112–132. **f** EM density (blue mesh) zoned 2 Å around a β-sheet comprising residues 181–188, 219–224 and 243–248.

rationalising the differences observed in the active site architecture[19] (Supplementary Fig. 5C).

The lectin domain jelly roll core is essentially conserved in HKU1, with RMSD values of 0.95 Å and 0.89 Å for BCoV and MCoV-NJ, respectively. However, whereas in BCoV and MCoVs extensive loops emanate from this jelly roll core to form the LD carbohydrate-binding site (CBS), such loops are either much shorter or absent in the HKU1 HE structure (Fig. 2e–g). As compared to the BCoV and MCoV HE, HKU1 HE underwent deletions in the β5–β6 loop, the β7–β8 loop, and β9–β10 loop (β11–β12 in BCoV) (Fig. 2 and Supplementary Fig. 6). The long β4–β5 loop has undergone a minor deletion in HKU1 HE but is also displaced with respect to that of BCoV and MCoV HE, reaching a maximum distance of 10 Å between residues 144 and 148. In BCoV and MCoV HE, each of these loops participates in ligand/receptor binding. In addition, the metal binding site (MBS) which stabilises the β11–β12 CBS loop in MCoV-NJ and BCoV (β13–β14) is absent in HKU1 (Fig. 2e–g). Two of the three metal coordinating sidechains, present in BCoV HE (Supplementary Fig. 7A), are partially conserved in HKU1, indicating that the MBS was likely present in the ancestral protein. However, the critical D220 site in BCoV HE falls within the β7–β8 loop deletion of HKU1 HE (Supplementary Fig. 7B). Thus, HKU1 has lost all

essential components of the HE CBS and the LD domain has been stripped back essentially to the jelly roll core. Following the historical nomenclature used for naming jelly roll beta strands (CHEF and BIDG), the deletions localise to the BC, DE, FG and HI loops at the thin end of the wedge-shaped beta sandwich (Supplementary Fig. 8A, B).

**Mapping of N-linked glycans by cryo-EM and glycoproteomics.** HKU1 HE contains eight predicted N-linked glycosylation sites which are strictly conserved between all HKU1 field strains studied so far. To better characterise the occupancy and composition of each of these N-linked glycosylation sites, we performed in-depth glycoproteomics profiling of the same recombinant HEK293T cell-derived material used for cryo-EM. HKU1 HE was digested in parallel with trypsin, chymotrypsin and alpha-lytic protease and analysed by reverse phase liquid chromatography coupled with tandem mass spectrometry. We used electron transfer high-energy collision dissociation (EThcD) fragmentation for the identification of N-linked glycopeptides (Supplementary Fig. 8). A total of 150 unique glycoforms were identified across all 8 N-linked glycosylation sites (Fig. 3a), 99 of which with sufficient signal for semi-quantitative analysis (Supplementary

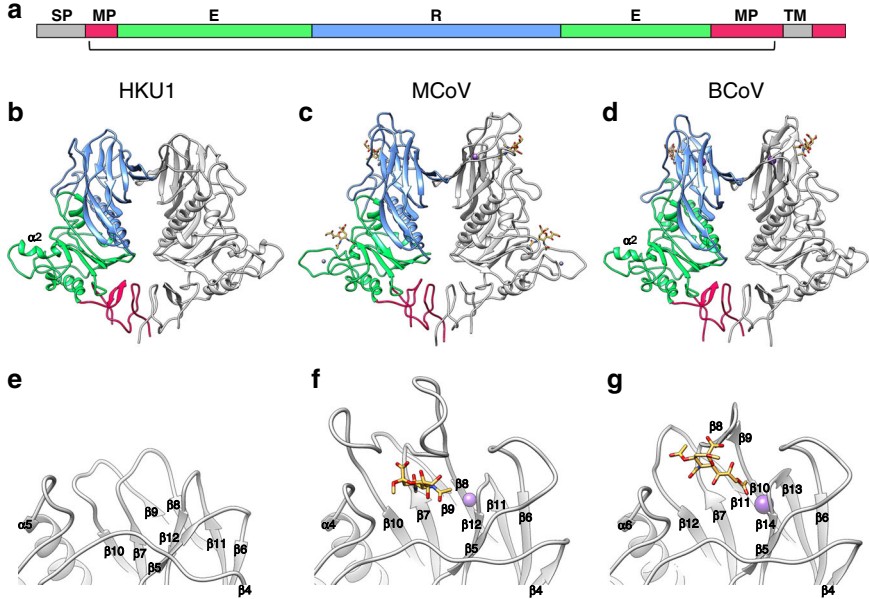

**Fig. 2 Comparison of HKU1 HE to related embecoviruses. a** Linear representation of HE domain organisation. The membrane-proximal domain (MPD), esterase domain (E) and receptor domain (R) are coloured red, green and blue, respectively. Grey segments indicate the signal- peptide (SP) and transmembrane (TM) domain. The bracket indicates the part of the protein for which the structure of HKU1 HE was solved. **b** Ribbon representation of the dimeric HKU1-A HE (residues 15–346), **c** MCoV-NJ HE (PDB ID: 5JIL, residues 20–388) and **d** BCoV-Mebus HE (PDB ID: 3CL5, residues 19–376) structures. One monomer is coloured grey, the other by domain organisation shown in (**a**). Bound 4-O-acetylated (MCoV) or 9-O-acetylated (BCoV) sialic acid is coloured orange. Sodium (MCoV) and potassium (BCoV) ions are coloured purple and zinc is grey. **e** Expanded view of the region corresponding to the LD carbohydrate-binding site (CBS) in HKU1 HE, **f** MCoV, and **g** BCoV.

Data 1). It should be noted that this semi-quantitative analysis does not account for potential differences in detection efficiency between the various glycoforms. A consolidated summary of the site-specific glycosylation profile by compositional class, fucosylation and sialylation, taken from selected protease datasets, is presented in Fig. 3a (a full overview of the trypsin, chymotrypsin and alpha-lytic protease datasets is presented in Supplementary Fig. 9, with glycoform-specific replicate analyses in Supplementary Data 1). As expected for materials derived from HEK293T and related cell-lines[27], glycosylation was predominantly of complex type and very heterogeneous, ranging from 8 to 59 unique glycoforms identified for each site (Fig. 3a). For sites N145, N168 and N193, situated on the LD loops (Fig. 3b), we also detected substantial signals for the unmodified asparagines, without glycosylation. Based on the combined signal intensities of all glycoforms, we found that the occupancy of those sites is ~81% for N145, <2% for N168 and 44% for N193. The low occupancy of N168 agrees with the lack of density observed in the cryo-EM map (Fig. 3c). Furthermore, the high B-factors of the LD loops suggests flexibility in this region (Supplementary Fig. 4E), which explains the limited density for N110, despite having 100% occupancy. With the exception of N168, the first core N-acetyl glucosamine (GlcNAc) was modelled for each of the LD loop glycans (Fig. 3c). The remaining four sites are fully occupied based on our MS data, in accordance with strong densities observed in the cryo-EM map. Indeed, we were able to model the entire $Man_3GlcNAc_2$ core for N286 (Fig. 3c). Apart from differences in glycan occupancy, we also observed marked differences in glycan composition. Whereas the overall pattern is dominated by complex glycosylation, sites N83 and N328 show predominant hybrid and high-mannose glycosylation, respectively. Sites N110 and N145, which contain mostly complex glycans, are also heavily (core) fucosylated and contain higher numbers of sialic acids. Whereas glycosylation varies substantially from site to site, and is very heterogeneous, we did identify a set

of glycan compositions that are highly abundant and shared at the majority of sites, as listed in Supplementary Table 3.

**Glycosylation of HKU1 HE lectin domain**. Side-by-side comparison of HKU1, MCoV-NJ, BCoV and OC43 reveals that HKU1 HE lectin domain has a much flatter topography (Fig. 4a–d). In addition to the dramatically altered LD, HKU1 has undergone substantial changes in the pattern of N-linked glycosylation in this region. Interestingly, the conserved N-linked glycan present at the LD dimer interface in MCoV-NJ (N241), BCoV (N236) and OC43 (N235) is lost in HKU1 (Fig. 4e–h). Compared to related embecoviruses, the LD of HKU1 has four unique N-linked glycosylation sites: N110, N145, N168 and N193, two of which localise to the remnants of the elongated β5-β6 loop and β7-β10 loop. The small β4-β5 loop deletion in HKU1 disrupts a conserved glycosylation site in OC43, BCoV and MCoV-DVIM (Supplementary Fig. 6). However, HKU1 reacquired an equivalently positioned glycan (N145) in the β4-β5 loop (Fig. 4a). Interestingly, HKU1 and OC43 have independently acquired a glycan adjacent to the former sialic acid binding site, N110 and N114, respectively (Fig. 4a, d). When viewed in the context of the dimeric structure, HKU1 appears to have distributed its N-linked glycans to the periphery of the lectin domain.

In an attempt to understand the evolutionary benefit of LD loop deletions and increased N-linked glycosylation, we first looked at the sequence conservation of BCoV HE. The BCoV LD exhibits modest sequence variation which localises primarily to the prominent β5–β6 loop, which participates in 9-O-Sia binding (Fig. 5a). In contrast, comparison of available HKU1-HE sequences reveals greater overall sequence variation on the vestigial LD (Fig. 5b). In the sharpened cryo-EM map, the glycans on the LD are not visible beyond the first core GlcNAc. However, mass spectrometry analysis confirms that these primarily contain complex glycans, comprising between 9 and 16 saccharide units.

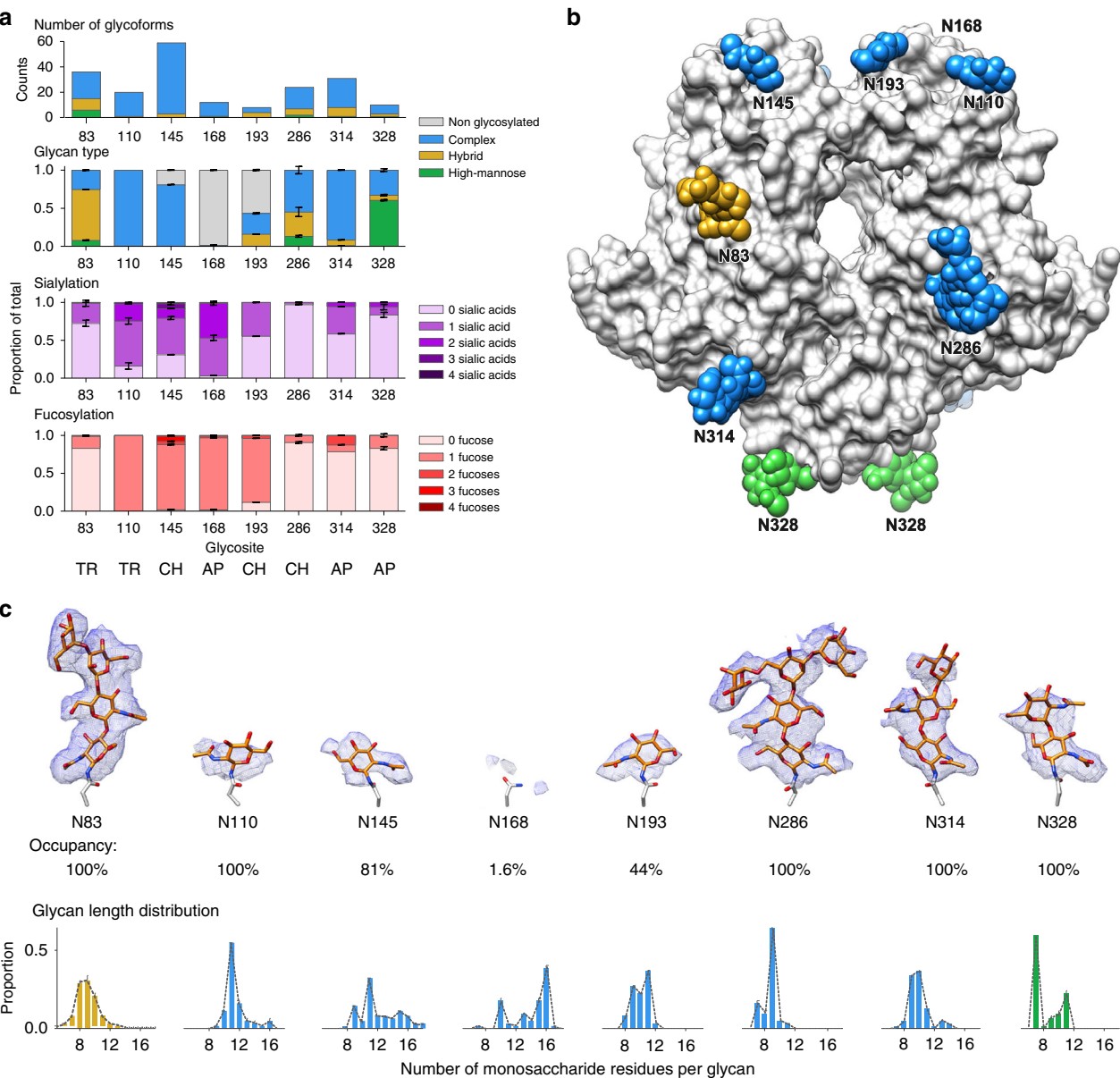

**Fig. 3 N-glycosylation of HKU1 HE. a** Glycoproteomic analysis of *N*-linked glycosylation in HKU1 HE. Top panel enumerates the total of unique glycan compositions identified per site. Bottom three panels show semiquantitative analyses from extracted peak areas of site-specific *N*-glycosylation by glycan type (non-glycosylated, high-mannose, hybrid, or complex), fucosylation, and sialylation. Error bars represent the standard deviation of the duplicate measurement. The selected protease dataset from which the data was extracted is indicated for each site: TR trypsin, CH chymotrypsin, AP alpha-lytic protease. A full overview is presented in Supplementary Fig. 9 and Supplementary Data 1. **b** Surface representation of the dimeric HKU1 HE atomic model, with modelled *N*-glycans shown as spheres and coloured according to the predominant glycan type shown in (**a**). **c** EM density (blue mesh) zoned 2 Å around each of the modelled *N*-glycans and analogous region for N168. The occupancy and glycan length distribution from glycoproteomics analysis for each site is shown below.

To understand where these lesser-ordered regions are situated, a difference map of the *N*-linked glycans was generated and a gaussian filter was applied in order to highlight low resolution features. Interestingly, the disordered portion of these glycans forms a crown of glycan density, which encircles the LD and covers much of its surface (Fig. 5c). Of note, density belonging to the N110 glycan of HKU1 overlaps with the former sialic acid binding site. Side-by-side comparison shows that the LD of HKU1 is ~8 Å shorter than BCoV, with none of the CBS loop remnants protruding above the glycan crown (Fig. 5d–f). While our mass spectrometry data reveals that the N168 site is only 1.6% occupied, we do observe difference density which extends

tangentially from this position. The esterase domain active site is highly conserved in both BCoV and HKU1 (Fig. 5d–f).

## Discussion

Coronaviruses pose a constant zoonotic threat, as poignantly illustrated by the ongoing SARS-CoV-2 pandemic[6]. It is, therefore, of relevance to understand how they cross-species barriers and subsequently adapt to their new hosts. Viruses in the sub-genus *Embecovirus*, a minor clade in the genus *Betacoronavirus*, seem particularly apt at crossing species barriers. Two of its members, OC43 and HKU1, arose from separate zoonotic introductions to become firmly established human pathogens[12].

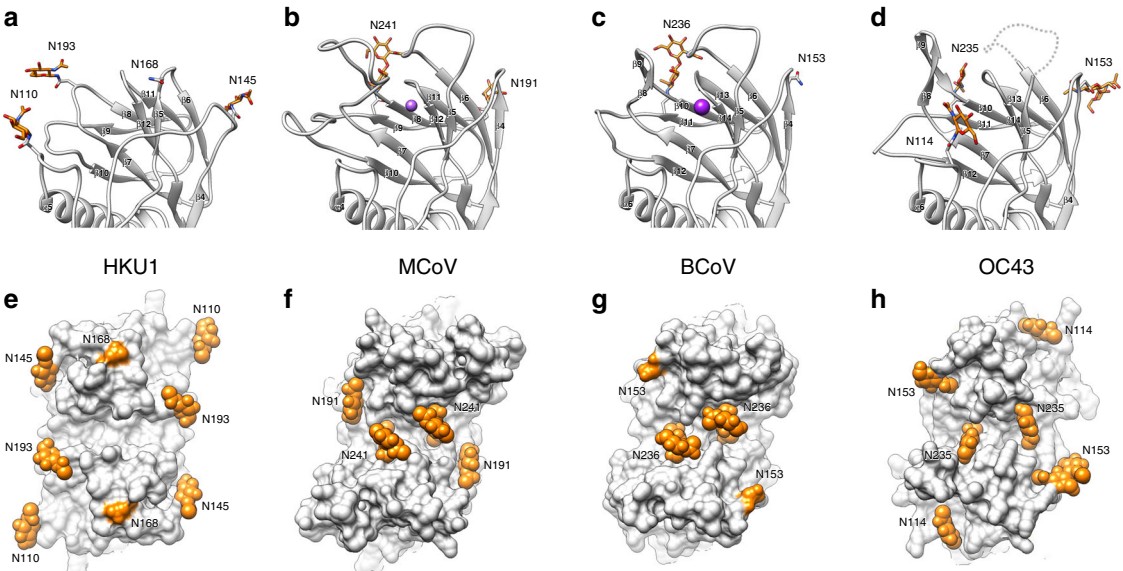

**Fig. 4 Comparison of HKU1 HE lectin domain glycosylation to related embecoviruses. a** Ribbon representation of the monomeric HE of HKU1, **b** MCoV-NJ, **c** BCoV and **d** OC43 lectin domain, with an approximation of the unmodelled β5-β6 loop shown as a dashed line. Each lectin domain glycosylation site is labelled and colour-coded according to their conservation across the four HEs shown. Modelled *N*-linked glycans are shown as sticks and coloured orange. **e** Top down view of the dimeric HE of HKU1 HE, **f** MCoV-NJ (PDB ID: 5JIL), **g** BCoV (PDB ID: 3CL5) and **h** OC43 (PDB ID: 5N11) lectin domain, shown as a surface representation. Modelled *N*-linked glycans, shown as spheres, and unmodelled *N*-glycosylation sites, are coloured orange and labelled as shown in (**a–d**).

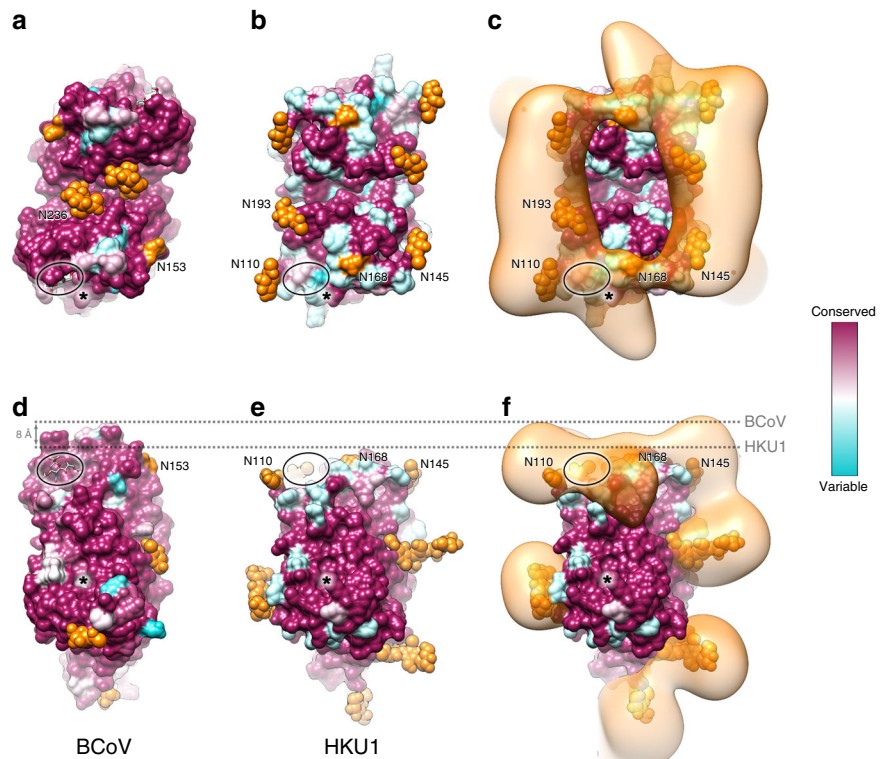

**Fig. 5 Visualizing the glycan crown of HKU1 HE lectin domain. a** Top down view of the dimeric BCoV HE lectin domain, shown as a surface representation and coloured according to sequence conservation (67 sequences). *N*-linked glycosylation sites are labelled, coloured orange and, if modelled, shown as spheres. The sialic acid binding site is circled and esterase active site is indicated by an asterisk. **b** Equivalent view as shown in A for HKU1 HE (40 sequences). **c** The same view as shown in B overlaid with the HKU1 HE *N*-linked glycan EM difference density, coloured orange. **d** Side view of BCoV HE and (**e–f**) HKU1 HE, depicted as in (**a–c**). The most membrane distal region of BCoV and HKU1 HE is indicated by a dashed line.

Their adaptation to replication in the human respiratory tract selected for loss of HE-mediated receptor-binding, which in turn downregulated HE receptor-destroying activity with consequences for dynamic virion-receptor interactions. In OC43, which emerged only recently[13], HE lectin function was reduced and ultimately lost through accumulation of single site mutations in LD[8]. HKU1 likely followed a similar evolutionary trajectory early upon its entry into the human population, but after prolonged circulation[11], the HE LD underwent far more consequential structural changes as revealed here by our cryo-EM analysis. Its overall structure is still highly similar to that of other CoV HEs, especially in the ED. Of the LD, the jelly roll core remains intact, but the CBS loops are almost completely gone and, concomitantly, the number of N-linked glycosylation sites increased. The data suggest that during HKU1 evolution, the LD was trimmed back substantially, once it became functionally obsolete. However, the core structure of the LD and the integrity of its fold are essential for the correct folding and function of the esterase domain and so it remains a functional module. The findings beg the question as to why the CBS loops were deleted. One possible explanation is that the protruding surface-exposed regions of the LD, once they lost their function in ligand binding, remained an antigenic liability. Indeed, embecovirus HEs are under selective pressure of the humoral immune response, as antibodies against HE neutralise BCoV and OC43 infectivity in vitro and confer protection against BCoV and MCoV in vivo[28–31]. The considerable sequence variation in the remaining surface exposed LD regions of HKU1 HE is suggestive of immune pressure[26,32].

Compared to HEs that retain their receptor-binding function, the LD of HKU1 HE exhibits increased N-linked glycosylation. There are eight N-linked glycosylation sites (Supplementary Fig. 10), strictly conserved among HKU1 HEs, four of which encircle the LD. Remarkably, however, the LD dimer interface glycan conserved in BCoV, MCoV and OC43, was lost in HKU1 (Fig. 4e–h). Conceivably, this dimer interface glycan could serve a structural role in stabilizing the CBS architecture, no longer required in HKU1 HE. Here we show that the site-specific glycan occupancy and composition varies widely from site to site, including those in the LD, in the recombinant materials used for these structural studies. Previous glycoproteomics analyses of coronavirus spike proteins have suggested that site-specific glycan processing and distribution of compositional classes are accurately represented in recombinant materials, compared to whole virions[33]. Whether this is also the case for HKU1 HE remains to be investigated, but this represents a significant challenge due to the difficulties of culturing HKU1 virions in sufficiently large quantities[34]. Our data suggest that in the context of recombinant expression in HEK293T cells, the HKU1 HE ectodomain structure poses several local restrictions to glycan occupancy and processing, resulting in a highly varied site-specific glycosylation pattern.

There are several possible explanations for the newly acquired N-linked glycans in the LD, including (i) with loss of the hydrophilic LD loops they add hydrophilicity and thereby aid folding of the hydrophobic core (Supplementary Fig. 11), (ii) they form a glycan shield, and/or (iii) some of them contributed to loss of receptor-binding function early on in HKU1 evolution. With respect to the latter option, the introduction of a glycan at N110 would predictably have sterically hindered receptor binding in the context of a functional LD, as its calculated difference density would have overlapped with the 9-O-Ac-Sia binding site. Indeed, in OC43 HE, the introduction of a glycosylation site at residue 114 in the same loop was one of four mutations acquired early after viral introduction in humans, and which contributed the most to loss of CBS affinity[8].

During circulation of OC43 in the last 70–120 years, the HE LD lost its capacity to bind 9-O-Ac-Sia through the accumulation of single site mutations, but its structure remained essentially intact. Over a longer timespan OC43 HE may well follow the fate of its HKU1 homologue, however, through similar deletions of the surface-exposed loops in the LD.

From a general evolutionary perspective, our data offers insights into how viruses deal with a structurally integral but functionally obsolete component of multidomain proteins. A number of parallels can be drawn between our observations of HKU1 HE and the haemagglutinin (HA) protein of influenza viruses. Firstly, the presence of a vestigial lectin domain in HKU1 HE is reminiscent of the vestigial esterase domain observed in the HA protein of influenza A and B viruses[35], albeit over a much shorter evolutionary period. In addition, similarly to the HKU1 and OC43 HEs, the HAs of both H1N1 and H3N2 influenza viruses have acquired and retained N-glycosylation sites since their introduction to humans, most of which localise to the variable globular head domain[36]. Finally, receptor binding site loop deletions have also been reported for H7N2 and H9N2 avian influenza HA. In both cases, sialic acid binding was retained but the receptor preference was altered. In the latter example, these deletions were shown to facilitate immune escape[37,38].

Our findings reinforce the notion that for embecoviruses, the HE, in addition to S, should be considered a viable candidate for vaccine development. Moreover, the highly conserved HE esterase active site offers an attractive target for the development of broad-spectrum antivirals against OC43 and HKU1 as well as against novel human embecoviruses, should such emerge in the future. By demonstrating that cryo-EM can be used to study CoV HEs at high-resolution, we expand the structural biology toolkit for future research, which may focus on antibody or inhibitor complexes.

## Methods

**Protein expression and purification.** The human codon-optimised sequence for HKU1 HE (Q5MQD1) was cloned into a pCD5-T-Fc expression plasmid[17]. The resulting construct encodes a chimeric protein comprising of the HE ectodomain fused to the human IgG1 Fc domain, with the domains separated by a thrombin cleavage site. The HE-Fc was produced by transient expression in HEK293T cells grown in 293 SFM II expression medium (Invitrogen) supplemented with 44.1 mM sodium bicarbonate, 11.1 mM glucose, Primatone RL-UF (3.0 g/liter), penicillin (100 IU/ml), streptomycin (100 μg/ml), 1% glutaMAX (Gibco), and 1.5% DMSO. The HE-Fc was then purified from cell culture supernatants by protein A affinity chromatography (GE Healthcare). The HE-Fc bound protein A beads were pelleted at $400 \times g$, washed four times with cleavage buffer (10 mM Tris-HCl pH 8.2 containing 50 mM NaCl), followed by on-bead thrombin (Sigma Aldrich) cleavage overnight at room temperature[17]. The next day, the beads were pelleted, the HE ectodomain in the supernatant was concentrated to a volume of ~150 μl and injected on a Superdex200 increase column (GE Healthcare) pre-equilibrated in 10 mM Tris-HCL pH 8 and 50 mM NaCl. Monodisperse dimeric fractions were concentrated to a final concentration of 4.0 mg/ml (Supplementary Fig. 1A). Sample purity was assessed by SDS-PAGE gel analysis (Supplementary Fig. 1B).

**Cryo-electron microscopy.** Three μl of purified HKU1 HE (4 mg/ml) was dispensed on Quantifoil R1.2/1.3 200-mesh grids (Quantifoil Micro Tools GmbH) that had been freshly glow discharged for 30 seconds at 20 mA using GloQube Glow Discharge system (Quorum Technologies). Grids were blotted for five seconds using Whatman No. 1 filter paper and immediately plunge-frozen into liquid ethane cooled by liquid nitrogen using a Vitrobot Mark IV plunger (Thermo Fisher Scientific) equilibrated to ~95% relative humidity, 4 °C. Movies of frozen-hydrated HKU1 HE were collected using Titan Krios G4 Cryo-TEM (Thermo Fisher Scientific) operating at 300 keV and equipped with a Falcon 4 Direct Electron Detector (Thermo Fisher Scientific). All cryo-EM data were acquired using the EPU 2 software (Thermo Fisher Scientific). Microscope was aligned to produce fringe-free imaging (FFI) allowing five acquisition areas within a hole and aberration-free image shift (AFIS) was used to acquire images from up to 21 holes per single stage move. Movies were collected in electron counting mode at 96,000× corresponding to a calibrated pixel size of 0.805 Å/pix over a defocus range of −1.0 to −2.5 μm. 6,029 movies were collected using a dose rate of 5 e−/pix/s for a total of 5.6 s (207 ms per fraction, 27 fractions), resulting in a total exposure of ~40 e−/Å (1.5 e−/Å²/fraction).

**Image processing**. Collected movie stacks were manually inspected and then imported in Relion version 3.0.1.[39] Drift and gain correction were performed with MotionCor2[40], and GCTF[41] was used to estimate the contrast transfer function for each movie. Movies with a GCTF-estimated resolution of 10 Å or worse were discarded. One thousand particles were picked manually and 2D classified. The resulting classes were then used as templates for autopicking in Relion, resulting in 936,260 particles. Fourier binned (2 × 2) particles were extracted in a 90 pixel box and subjected to a round of 2D classification after which 565,194 particles were retained. Using the 'molmap' command in UCSF chimera[42], the HKU1 HE homology model was used to generate a 40 Å resolution starting model for 3D classification. Particles selected from 2D classification were subject to a round of 3D classification (with C2 symmetry). Particles belonging to the best class (119,717 particles) were re-extracted unbinned in a 320 pixel box to ensure delocalised high-resolution information was not excluded. Subsequent 3D auto-refinement (with C2 symmetry) and post-processing yielded a map with a resolution of 3.79 Å. Further sub-classification attempts did not lead to improvements in map quality or resolution. Per particle defocus estimation improved the resolution to 3.73 Å. Relion's Bayesian polishing procedure was then performed on these particles, with all movie frames included, which produced a 3.68 Å map[43]. Next, particles were assigned to their respective AFIS group and subject to beam-tilt refinement, which further improved the resolution to 3.48 Å. Data processing were then continued in Relion version 3.1b[44], where a final round of per-particle defocus and per-micrograph astigmatism estimation was performed. This produced a 3.39 Å resolution map, based on the gold-standard FSC = 0.143 criterion. A negative B-factor of −122 Å$^2$ was applied during the final post-processing step. Local resolution estimations were performed using Relion. An overview of the data processing pipeline is shown in Supplementary Fig. 2.

**Model building and refinement**. Initially, a homology model of HKU1 HE (uniprot ID: Q5MQD1) was generated using the phyre2 server[45]. Each protomer was individually fitted in EM density map using the UCSF Chimera 'fit in map' tool[42]. The resulting dimeric model was then edited in Coot using the 'real space refinement', carbohydrate module and 'sphere refinement' tools[46,47]. Iterative rounds of manual fitting in Coot and real space refinement in Phenix[48] were carried out to improve non-ideal rotamers, bond angles and Ramachandran outliers. During real space refinement, secondary structure and NCS restraints were imposed. Validation was carried out using Molprobity (general/protein[49]) and Privateer[50,51]. All data collection, image processing and refinement information can be found in Supplementary Table 1.

**Analysis and visualisation**. The N-linked glycan difference map was generated in UCSF chimera[42]. Firstly, the 'molmap' command was used to generate a 4 Å resolution density map from the fitted protein-only atomic coordinates of the HKU1 HE dimer. This simulated map was then resampled on the grid of the experimental cryo-EM density map using the 'vop resample' command. Subsequently, a gaussian filter (σ = 4) was applied to the simulated and experimental map using the 'volume filter' tool. The 'vop subtract' command was then used to subtract the value of the simulated map from the experimental map. The 'minRMS' option was used to automatically scale the simulated map to minimize the root-mean-square sum of the resulting subtracted values at grid points within the lowest contour of the simulated map. The location of the resulting difference density agreed with the location of the modelled N-linked glycans, and disordered N and C-terminal regions located at the membrane-proximal region. A similar difference map could be obtained by using the 'color zone' tool to erase cryo-EM density within 3 Å of a fitted protein-only atomic coordinates and then applying a gaussian filter (σ = 4) to the remaining density. Multiple sequence alignments of BCoV and HKU1 HE were generated using NCBI BLAST and subsequently plotted onto their respective structures using UCSF Chimera (Supplementary Data 2). Alignment of the representative HKU1 genotype A and B sequences (Supplementary Fig. 10) was generated using the EMBL-EBI Clustal Omega programme[52]. Surface colouring of BCoV and HKU1 HE using the Kyte-Doolittle hydrophobicity scale was performed in UCSF chimera. RMSD values were calculated using the 'MatchMaker' tool of UCSF Chimera with default settings. Figures were generated using UCSF Chimera[42], UCSF ChimeraX[53] and PyMOL (The PyMOL Molecular Graphics System, Version 2.0, Schrödinger, LLC).

**Sample preparation for glycoproteomics analysis**. Nine μg of CoV-HKU1 HE were incubated in 100 mM Tris pH 8.5, 2% sodium deoxycholate, 10 mM tris(2-carboxyethyl)phosphine, and 40 mM iodoacetamide at 95 °C for ten minutes and at 25 °C for 30 min in the dark. Denatured, reduced and alkylated CoV-HKU1 HE (3 μg) was then diluted into fresh 50 mM ammonium bicarbonate and incubated overnight at 37 °C either with 0.056 μg of trypsin (Promega), chymotrypsin (Sigma Aldrich) or alpha-lytic protease (Sigma Aldrich). Formic acid was then added to a final concentration of 2% and the samples were centrifuged at 21,100 × g for 20 min at 4 °C, followed by another round of centrifugation for 5 min to precipitate the sodium deoxycholate and collect the peptides from the supernatants. Then, the CoV-HKU1 HE tryptic, chymotryptic and alpha lytic protease digests were desalted using 30 μm Oasis HLB 96-well plate (Waters). The Oasis HLB sorbent was activated with 100% acetonitrile and subsequently equilibrated with 10% formic acid in water. Next, peptides were bound to the sorbent, washed twice with 10% formic acid in water and eluted with 100 μL of 50% acetonitrile/5%formic acid (v/v). The eluted peptides were vacuum-dried and resuspended in 100 μL of 2% formic acid in water.

**Mass spectrometry**. Six μl of resuspended peptides for the glycoform identification and two μl two-fold diluted peptides for the glycoform quantification were run on an Orbitrap Fusion Tribrid (ThermoFisher Scientific, Bremen) mass spectrometer coupled to nanospray UHPLC system Agilent 1290 (Agilent Technologies) in duplicates. A 90-min LC gradient from 0% to 35% acetonitrile was used to separate peptides at a flow rate of 300 nl/min. A Poroshell 120 EC C18 (50 cm × 75 μm, 2.7 μm, Agilent Technologies) analytical column and a ReproSil-Pur C18 (2 cm × 100 μm, 3 μm, Dr. Maisch) trap column were used for the peptide separation. The data were acquired in data-dependent mode. Orbitrap Fusion parameters for the full scan MS spectra were as follows: an AGC target of 4 × 10$^5$ at 60,000 resolution, scan range 350–2000 m/z, Orbitrap maximum injection time 50 ms. Ten most intense ions (2+ to 8+ ions) were subjected to fragmentation with electron-transfer/higher energy collision dissociation ion fragmentation scheme[54]. The supplemental higher energy collision dissociation energy was set at 30%. The MS2 spectra were acquired at a resolution of 30,000 with an AGC target of 5∗10$^5$, maximum injection time 250 ms, scan range 120–4000 m/z and dynamic exclusion of 16 s.

**Mass spectrometry data analysis**. The acquired data were analysed using Byonic[55] against a custom database of recombinant CoV-HKU1 HE protein and used proteases, searching for glycan modifications with 12/24 ppm search windows for MS1/MS2, respectively, and a False Discovery Rate (FDR) set to 1%. Up to three missed cleavages were permitted with C-terminal cleavage at R/K for trypsin, F/Y/W/M/L for chymotrypsin and T/A/S/V for alpha-lytic protease. Carbamidomethylation of cysteine was set as fixed modification, methionine oxidation as variable common 1, glycan modifications as variable common 2, allowing up to max. 2 variable common parameters per one peptide. A glycan database containing 305 N-linked glycans was used in the search. Glycopeptide hits reported in the Byonic results file were initially accepted if the Byonic score was ≥200, LogProb ≥2, and peptide length was at least 6 amino acids. Accepted glycopeptides were manually inspected for quality of fragment assignments. The glycopeptide was considered true-positive if the appropriate b, y, c and z fragment ions were matched in the spectrum, as well as the corresponding oxonium ions of the identified glycans. All glycopeptide identifications were merged into a single non-redundant list per sequon. Glycans were classified based on HexNAc content as high-mannose (2 HexNAc), hybrid (3 HexNAc) or complex (>3 HexNAc). Byonic search results were exported to mzIdentML format to build a spectral library in Skyline[56] and extract peak areas for individual glycoforms from MS1 scans. The full database of variable N-linked glycan modifications from Byonic was manually added to the Skyline project file in XML format. Glycopeptide identifications from Byonic were manually inspected in Skyline and evaluated for correct isotope assignments and well-defined elution profiles, suitable for peak integration. In the case of multiple missed cleavages, reporting on the same site-specific glycoform, peak areas were summed in the semi-quantitative analysis. Reported peak areas were pooled based on the number of HexNAc, Fuc or NeuAc residues to distinguish high-mannose/hybrid/complex glycosylation, fucosylation and sialylation, respectively. The semi-quantitative analysis of the glycosylation profile was performed per site, per protease. For the data presented in Fig. 3a, protease datasets were selected based on coverage and overall signal for the corresponding glycosylation site. The quantified data were represented with GraphPad Prism 8 software.

**Reporting summary**. Further information on research design is available in the Nature Research Reporting Summary linked to this article.

## Data availability

Coordinates are deposited in the Protein Data Bank under accession code 6Y3Y. The corresponding EM density maps (final unsharpened, sharpened, local resolution filtered, half maps, N-linked glycan difference map, and mask) have been deposited to the Electron Microscopy Data Bank under the accession EMD-10676. The unaligned gain-normalised movies are available on the Electron Microscopy Public Image Archive under the accession EMPIAR-10390. The raw LC-MS/MS files and glycoproteomics analyses have been deposited to the ProteomeXchange Consortium via the PRIDE partner repository with the dataset identifier PXD017545. All reagents and relevant data are available from the authors upon request.

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

## Acknowledgements

D.L.H. is funded from the European Union's Horizon 2020 research and innovation program under the Marie Skłodowska-Curie grant agreement (No. 842333) and holds an EMBO non-stipendiary long-term Fellowship (ALTF 1172-2018). Research reported in this publication was supported by a China Scholarship Council grant to Y.L. (No 2014-

03250042). T.M.S., M.F.P., and J.S. are supported by the Dutch Research Council NWO Gravitation 2013 BOO, Institute for Chemical Immunology (ICI, 024.002.009). We thank Dr. Mihajlo Vanevic for IT support and Dr. Lingbo Yu for her assistance processing AFIS data.

## Author contributions

D.L.H., Y.L., and R.J.G. conceived the project. Y.L. generated the HKU1 HE expression construct and expressed the protein. D.L.H. and Y.L. performed protein purification. T.M.S. and J.S. performed glycoproteomic experiments and analysed the data. I.D. prepared cryo-EM grids and set up data collection. D.L.H. carried out single-particle image processing. D.L.H. and M.F.P. built and analysed the structure. D.L.H., J.S., F.J.M.K., and R.J.G. supervised the project. D.L.H., J.S. and R.J.G. wrote the first draft and all authors contributed to the preparation of the final paper.

## Competing interests

I.D. is an employee of Thermo Fisher Scientific. The other authors declare no competing interests.
