## [Peer Review File · Nature Communications]

REVIEWER COMMENTS

Reviewer #1 (Remarks to the Author):

Comprehensive structural characterization of viral proteins is important for the development of vaccines and antivirals, which is particularly relevant for zoonotic viruses. Hurdiss et al. use state-of-the-art cryo-EM and mass spectrometry technologies to resolve the structure and determine site-specific glycoprofiles on human betacoronavirus HKU1 hemagglutinin esterase (HE), which is refractory to crystallization methods. The authors demonstrate that the structure of HKU1 HE lectin domain is substantially different from those of related animal CoV HEs, resulting in loss of sialic acid binding activity and likely altering sialoglycan engagement dynamics of virions *in vivo*. The study therefore offers structural insight into *Embecovirus* adaptation to human host. The authors also suggest that over evolution HKU1 HE acquired N-glycosites to cover up the non-functional truncated lectin domain, though this is yet to be demonstrated on infectious virions, particularly in relation to types of glycan structures. While this is an elegant and relevant study, I miss a more thorough description of the MS data analysis and discussion of potential pitfalls. Therefore, some specific issues should be addressed.

1. The bar graphs in figure 3A represent proportions of glycan structures of different compositional classes, but there is no estimate of variation. Please include error bars or dot plots for each replicate.

2. While the MS-related data have been deposited on PRIDE servers, it only contains analyses related to identification, aside from the raw files. There is too little detail on how the quantification data was analyzed in Table S2. How were the different protease cleavage products (with and without missed cleavages) combined to come to the final result in figure 3A? What was considered as "sufficient signal"? The data analysis should be described in more detail in the method section and the supplementary table.

3. The authors do not consider differences in ionization efficiencies for non-modified vs glycosylated peptides, which may overestimate the proportion of non-glycosylated peptides. This should be discussed.

4. In figure 4, are the modelled glycans for the other strains based on pre-existing experimental data? Or are the glycans modelled based on the consensus sequons in the lectin domain? If the latter is true, additional glycan should be modelled for BCoV next to the β 4- β 5 loop, which would roughly (2 aa away) correspond to N145 of HKU1.

5. The authors do not reflect on any potential differences between the glycosylation of recombinant HE and native HE on viral particles. Substantial differences of site-specific N-glycan structures have been identified on recombinant vs native viral proteins, which may be related to glycosylation capacity of expression cell line, as well as accessibility to glycosyltransferases and secretory pathway transition time of a soluble construct. The reference to expected HEK293T cell glycoprofile is not entirely correct, as HEK293 and not HEK293T cells are analyzed in the cited paper. Though there are substantial similarities, the two cell lines vary in levels of sialylation and fucosylation.

Minor:

1. Line 151. N286 and not N314 based on figure.

2. Figure S5. Please include the OC43 HE in the multiple sequence alignment.

3. Figure 5. Please include a supplementary file listing the 75 strains of BCoV and the 40 strains of HKU1 used for generating the figure.

Reviewer #2 (Remarks to the Author):

This article by Hurdiss et al presents an intriguing, well-rounded story covering the architectural evolution of the haemagglutinin esterase of coronavirus-HKU1. It is well written and well presented, although I would like to propose some changes that might make it more accessible to a broad readership. Their contribution is timely on two fronts: one, due to the current COVID-19 pandemic, any insight into whatever comes next after zoonosis is likely to help prepare for the future; two, there is a technological and methodological revolution going on on the Electron Cryo-microscopy front, and this article delivers another welcome push in the struggle to reconstruct ever smaller particles.

- My first point relates to the fact that an evolutionary line is being drawn but no phylogenetics are being shown. The article is very structural (nothing wrong about that) but I believe it is missing a graphical link to the big picture.

- In Fig. 3 and the related section of the main text, I have missed a comparison between the glycoform compositions found in HKU1 HE and those found on related coronaviruses. Are we seeing more variability here or elsewhere?

- I think the section on the difference map needs a bit more technical explanation, or more references where people can find more information about it.

- My only real criticism is towards the claim that Cryo-EM makes protein glycosylation more tractable than X-ray Crystallography. I do not think I am seeing any major breakthrough here in that respect? It is not unusual for the first four to five monosaccharides to be resolved in both electron density and electron potential maps. There does seem to be a difference in how glycans are being treated by the two communities though, mainly in terms of how much care is put into refining them (see <https://www.sciencedirect.com/science/article/abs/pii/S0959440X19301411> for a comparative graph).

Jon Agirre
University of York

Reviewer #3 (Remarks to the Author):

The paper presents the cryo-EM structure of the hemagglutinin-esterase (HE) surface glycoprotein of the coronavirus HKU1, a virus that infects humans. The protein is highly similar to the HE's from other coronaviruses but with important differences. The main finding is that while the esterase domain (which destroys receptors containing 9-OAc sialic acid) is intact and resembles, for example, bovine coronavirus HE, the structure of the lectin domain which in other cases functions in receptor binding shows that it retains a jellyroll topology but has lost the carbohydrate binding loops, thus explaining at the molecular level the loss of lectin activity. Another observation in the structure is the presence of N-linked carbohydrates on the lectin domain, not typically present on other HE's. Interestingly, the moiety at N110 actually blocks the sialic acid binding site, consistent with disuse of the lectin domain function. The authors suggest the carbohydrates act as a glycan shield against immune recognition of the function-less domain.

This study adds to our understanding of how multifunctional viral glycoproteins evolve and how their different activities are balanced with each other. The glycoproteins recognize receptors and therefore determine the host range specificity. For the case of sars-cov-2, which as we know has recently jumped species, the virus recognizes receptors through the S-protein and there is no HE

protein. In the virus studied here, the HE is combined with the S protein, which can also function in receptor binding.

The map appears to be at a decent resolution for model building and is an achievement for cryo-EM study of a small protein of 80 kDa covered with glycans. Additionally, the study uses mass spec to analyze the types of N-linked carbohydrate and their occupancies which gives more completeness to the carbohydrate assessment than available from the cryo-EM map. The paper is clearly written and will be of interest to general readers.

A structure of an HE with a disused lectin domain from coronavirus OC43 was studied previously by X-ray crystallography. I think the present paper will be improved by a brief comparative analysis of the structural changes that occurred in the HKU1 lectin domain compared to OC43. HKU1 has lost carbohydrate binding loops, but in Figure 4A and D, HKU1 looks quite similar to OC43. At the structural level are the cases similar or different?

In reporting the structure determination, a measure of agreement between the map and model should be provided such as the map-model FSC because good geometry as reported can be achieved without reference to the map. I note that an EMRinger score is provided. The authors should confirm that their starting model for 3D classification was low-pass filtered to 40 Å.

Reviewer #4 (Remarks to the Author):

The manuscript from Hurdiss et al reports the 3.4 Å cryo-EM structure and analysis of the HKU1-CoV haemagglutinin esterase (HE) ectodomain. The structure reveals that loops in the lectin domain have been truncated and are no longer capable of binding to glycans, possibly as a result of adaptation to replication in humans. The esterase domain (ED) is intact and is similar to other coronavirus EDs. The authors also perform a mass spec analysis of the N-linked glycosylation, which provides quantitative information of the glycan shield. It appears that the HKU1 HE lectin domain has acquired additional glycosylation sites in the truncated loops, perhaps to shield this region from recognition by the humoral immune system.

The manuscript is clearly written and the figures are excellent. Obtaining a 3.4Å cryo-EM reconstruction on a relatively small (~80 kDa) protein is commendable, and the map and model stats are very good. A weakness is that the structure did not provide much new information that could not have been obtained from a homology model derived from the HKU1 HE sequence and related HE structures from MHV and BCoV, but the structural analysis and the insights into evolution of the HE protein were interesting.

Reviewer #1 (Remarks to the Author):

Comprehensive structural characterization of viral proteins is important for the development of vaccines and antivirals, which is particularly relevant for zoonotic viruses. Hurdiss et al. use state-of-the-art cryo-EM and mass spectrometry technologies to resolve the structure and determine site-specific glycoprofiles on human betacoronavirus HKU1 hemagglutinin esterase (HE), which is refractory to crystallization methods. The authors demonstrate that the structure of HKU1 HE lectin domain is substantially different from those of related animal CoV HEs, resulting in loss of sialic acid binding activity and likely altering sialoglycan engagement dynamics of virions in vivo. The study therefore offers structural insight into Embecovirus adaptation to human host. The authors also suggest that over evolution HKU1 HE acquired N-glycosites to cover up the non-functional truncated lectin domain, though this is yet to be demonstrated on infectious virions, particularly in relation to types of glycan structures. While this is an elegant and relevant study, I miss a more thorough description of the MS data analysis and discussion of potential pitfalls. Therefore, some specific issues should be addressed.

1. The bar graphs in figure 3A represent proportions of glycan structures of different compositional classes, but there is no estimate of variation. Please include error bars or dot plots for each replicate.

We thank the reviewer for pointing out this oversight. Whereas the underlying replicate data was presented in Supplementary Table S2, this was not properly represented in the main text figure. Error bars have now been added to the figure in the revised manuscript.

2. While the MS-related data have been deposited on PRIDE servers, it only contains analyses related to identification, aside from the raw files. There is too little detail on how the quantification data was analyzed in Table S2. How were the different protease cleavage products (with and without missed cleavages) combined to come to the final result in figure 3A? What was considered as “sufficient signal”? The data analysis should be described in more detail in the method section and the supplementary table.

To improve the transparency of our quantitative analysis, we have added several clarifying statements to the methods section of the manuscript and extra information in supplementary table S2. Moreover, the Skyline project files used to extract the peak areas have been added to the PRIDE repository. To address the specific question of the reviewer here: when the same glycoform was detected on multiple peptides resulting from missed cleavages, the areas were summed. The quantitative analysis was performed per site, per protease. Whereas not all sites are covered by all proteases, we observe good agreement in the quantitation when sites are covered in multiple protease datasets (see Supplementary Table S2). The data in Figure 3A represent the consolidated results from multiple proteases. This choice of protease dataset is now further clarified by adding a label to each site in Figure 3A. A more extended version of Figure 3A, including the quantitation for all proteases for all sites is now included as an additional supplementary figure S9 in the revised manuscript. What constituted “sufficient signal for quantitation” was decided on a case-by-case basis, judging by the quality of the chromatographic profile. In practice, we included glycopeptides with peak areas ranging from 2e5 to 5e10.

3. The authors do not consider differences in ionization efficiencies for non-modified vs glycosylated peptides, which may overestimate the proportion of non-glycosylated peptides. This should be discussed.

Indeed, our MS experiments and quantitation do not account for potential differences in ionization/detection efficiency for differently modified forms of the peptide. We have added the clarifying statement: "It should be noted that this semi-quantitative analysis does not account for differences in detection efficiency between the various glycoforms."

That being said, we would like to point out to the reviewer that past work from the BioMS group in Utrecht has demonstrated repeatedly that this label-free bottom-up glycoproteomics method for relative quantitation provides robust glycan occupancy profiles for N-, O- and C-linked glycosylation, consistent with orthogonal methods of quantitation:

- Reiding, Karli R., et al. "Neutrophil myeloperoxidase harbors distinct site-specific peculiarities in its glycosylation." *Journal of Biological Chemistry* 294.52 (2019): 20233-20245.
- Franc, Vojtech, Yang Yang, and Albert JR Heck. "Proteoform profile mapping of the human serum complement component C9 revealing unexpected new features of N-, O-, and C-glycosylation." *Analytical chemistry* 89.6 (2017): 3483-3491.
- Yang, Yang, et al. "Hybrid mass spectrometry approaches in glycoprotein analysis and their usage in scoring biosimilarity." *Nature communications* 7.1 (2016): 1-10.
- Franc, Vojtech, Jing Zhu, and Albert JR Heck. "Comprehensive proteoform characterization of plasma complement component C8 $\alpha\beta\gamma$ by hybrid mass spectrometry approaches." *Journal of the American Society for Mass Spectrometry* 29.6 (2018): 1099-1110.

We are therefore confident that big differences in the relative intensities of unmodified peptides between the different sites in our glycoproteomics analyses reflect true differences in glycan occupancy. This is also reflected in the fact that the lowest occupied site in our glycoproteomics data, N168, shows a clear lack of density in the cryoEM reconstruction, even of the core GlcNAc residues.

4. In figure 4, are the modelled glycans for the other strains based on pre-existing experimental data? Or are the glycans modelled based on the consensus sequons in the lectin domain? If the latter is true, additional glycan should be modelled for BCoV next to the β 4- β 5 loop, which would roughly (2 aa away) correspond to N145 of HKU1.

The glycans shown in figure 4B-D and 4F-H correspond to what has been modelled in crystal structures (PDB IDs are provided in the figure legend). However, the reviewer quite rightly points out that BCoV has an N-glycosylation site at position N153 (although no glycan density is visible in the electron density map). While we initially decided to compare equivalently positioned glycans, upon reflection, we decided that comparing the absolute conservation of consensus sequons in the lectin domain adds more to the story (without changing the overall

conclusions). Lines 166-177 have been updated to reflect these changes and we have indicated the position of BCoV residue N153 in Figures 4, 5 and S11. The corresponding figure legends have also been updated. We thank the reviewer for prompting these changes which we believe have improved the manuscript.

5. The authors do not reflect on any potential differences between the glycosylation of recombinant HE and native HE on viral particles. Substantial differences of site-specific N-glycan structures have been identified on recombinant vs native viral proteins, which may be related to glycosylation capacity of expression cell line, as well as accessibility to glycosyltransferases and secretory pathway transition time of a soluble construct. The reference to expected HEK293T cell glycoprofile is not entirely correct, as HEK293 and not HEK293T cells are analyzed in the cited paper. Though there are substantial similarities, the two cell lines vary in levels of sialylation and fucosylation.

The reviewer rightly points out that our analyses were performed on recombinant HE, which may differ from native HE on viral particles. Likewise, glycosylation may vary depending on the specific cell type from which the viral particles – or recombinant protein, for that matter - were derived.

Glycosylation profiling on native viral particles poses additional challenges, as glycopeptides have to be detected and quantified against a background of other, often highly abundant, viral proteins. Despite these challenges, this profiling of has been successfully demonstrated on coronavirus spike proteins previously by us and the Crispin lab:

- Walls, Alexandra C., et al. "Unexpected receptor functional mimicry elucidates activation of coronavirus fusion." *Cell* 176.5 (2019): 1026-1039.
- Yao, Hangping et al. "Molecular architecture of the SARS-CoV2 virus." *bioRxiv* (2020) <https://doi.org/10.1101/2020.07.08.192104>

Whereas subtle differences may indeed be apparent between recombinant spike and native viral particles, the site-specific glycosylation patterns are actually remarkably similar. The distribution of compositional classes and degree of processing are virtually indistinguishable between recombinant and native spikes. Such similarities remain to be verified on a case-by-case basis, but these two studies both suggest that important aspects of the glycosylation patterns are accurately represented in the recombinant materials. And regardless of potential differences between recombinant and native HE, our glycoproteomics provide an accurate description of the materials that we used for our cryoEM studies, to aid in interpretation of the observed densities.

Moreover, HKU1 is notoriously difficult to culture, such that there is little prospect of profiling HE glycosylation on native particles. As it stands, our glycoproteomics experiments represent the sole investigation into HKU1 HE glycosylation to date (or any coronavirus HE, for that matter). A brief discussion along these lines has now been added to discussion section of the revised manuscript.

As for the reference to expected HEK293 glycosylation, the original manuscript included the qualifying statement that this referred to the degree of complex glycosylation and

heterogeneity. Whereas subtle differences in sialylation/fucosylation between HEK293, HEK293T, HEK293F etc. may indeed occur, this degree of complex glycosylation and heterogeneity remains typical of this 'family' of cell lines. This is now further clarified in the revised manuscript.

Minor:

1. Line 151. N286 and not N314 based on figure.

This has now been corrected (line 157).

2. Figure S5. Please include the OC43 HE in the multiple sequence alignment.

OC43 has now been included in a new multiple sequence alignment figure S6, generated using ESPript. We have also updated the HKU1-A and HKU1-B sequence alignment, figure S10, for stylistic consistency.

3. Figure 5. Please include a supplementary file listing the 75 strains of BCoV and the 40 strains of HKU1 used for generating the figure.

A supplementary excel file containing these sequences has been added (table S4). The actual number of BCoV HE sequences used is 67, but was erroneously stated as being 75. This has now been corrected in the figure 5 legend.

Reviewer #2 (Remarks to the Author):

This article by Hurdiss et al presents an intriguing, well-rounded story covering the architectural evolution of the haemagglutinin esterase of coronavirus-HKU1. It is well written and well presented, although I would like to propose some changes that might make it more accessible to a broad readership. Their contribution is timely on two fronts: one, due to the current COVID-19 pandemic, any insight into whatever comes next after zoonosis is likely to help prepare for the future; two, there is a technological and methodological revolution going on on the Electron Cryo-microscopy front, and this article delivers another welcome push in the struggle to reconstruct ever smaller particles.

- My first point relates to the fact that an evolutionary line is being drawn but no phylogenetics are being shown. The article is very structural (nothing wrong about that) but I believe it is missing a graphical link to the big picture.

We thank the reviewer for this suggestion. A phylogenetic tree depicting the evolutionary relationship between embecovirus HE has been added to the supplementary information as figure S1.

- In Fig. 3 and the related section of the main text, I have missed a comparison between the glycoform compositions found in HKU1 HE and those found on related coronaviruses. Are we seeing more variability here or elsewhere?

We agree that it would be interesting to compare the glycoforms present in HKU1 HE to those of OC43 or animal embecoviruses. While a number of studies have investigated the site-specific N-glycosylation of coronavirus spike proteins, this is not the case for HE. Thus, the present study is the first, to our knowledge, to describe such information for a coronavirus HE protein.

- I think the section on the difference map needs a bit more technical explanation, or more references where people can find more information about it.

We have updated the methods section with a more in-depth explanation of how the difference density can be generated from two similar approaches (lines 335-349).

- My only real criticism is towards the claim that Cryo-EM makes protein glycosylation more tractable than X-ray Crystallography. I do not think I am seeing any major breakthrough here in that respect? It is not unusual for the first four to five monosaccharides to be resolved in both electron density and electron potential maps. There does seem to be a difference in how glycans are being treated by the two communities though, mainly in terms of how much care is put into refining them (see <https://www.sciencedirect.com/science/article/abs/pii/S0959440X19301411> for a comparative graph).

We assume that this comment refers to the final line of the abstract "...that are intractable to X-ray crystallography". It was not our intention to claim that cryo-EM makes protein glycosylation more tractable than X-ray Crystallography. The point we are trying to convey is that researchers should not be afraid to employ cryo-EM to study heavily glycosylated small proteins, especially when X-ray crystallography isn't a viable option (such as with HKU1 HE). We have now removed the aforementioned statement to prevent this from being misconstrued. The final sentence of the abstract now reads "The findings further our insight into the evolution and host adaptation of human embecoviruses and also demonstrate the utility of cryo-EM for studying small, heavily glycosylated proteins."

Jon Agirre
University of York

Reviewer #3 (Remarks to the Author):

The paper presents the cryo-EM structure of the hemagglutinin-esterase (HE) surface glycoprotein of the coronavirus HKU1, a virus that infects humans. The protein is highly similar to the HE's from other coronaviruses but with important differences. The main finding is that while the esterase domain (which destroys receptors containing 9-OAc sialic acid) is intact and resembles, for example, bovine coronavirus HE, the structure of the lectin domain which in other cases functions in receptor binding shows that it retains a jellyroll topology but has lost the carbohydrate binding loops, thus explaining at the molecular level the loss of lectin activity. Another observation in the structure is the presence of N-linked carbohydrates on the lectin domain, not typically present on other HE's. Interestingly, the moiety at N110

actually blocks the sialic acid binding site, consistent with disuse of the lectin domain function. The authors suggest the carbohydrates act as a glycan shield against immune recognition of the function-less domain.

This study adds to our understanding of how multifunctional viral glycoproteins evolve and how their different activities are balanced with each other. The glycoproteins recognize receptors and therefore determine the host range specificity. For the case of sars-cov-2, which as we know has recently jumped species, the virus recognizes receptors through the S-protein and there is no HE protein. In the virus studied here, the HE is combined with the S protein, which can also function in receptor binding.

The map appears to be at a decent resolution for model building and is an achievement for cryo-EM study of a small protein of 80 kDa covered with glycans. Additionally, the study uses mass spec to analyze the types of N-linked carbohydrate and their occupancies which gives more completeness to the carbohydrate assessment than available from the cryo-EM map. The paper is clearly written and will be of interest to general readers.

A structure of an HE with a disused lectin domain from coronavirus OC43 was studied previously by X-ray crystallography. I think the present paper will be improved by a brief comparative analysis of the structural changes that occurred in the HKU1 lectin domain compared to OC43. HKU1 has lost carbohydrate binding loops, but in Figure 4A and D, HKU1 looks quite similar to OC43. At the structural level are the cases similar or different?

We thank the reviewer for pointing this out. The HE of OC43 and BCoV-Mebus are 95.1% identical (96.6% across their entire genome). As you would expect, the structure of their HE is also very similar, although this similarity is not well depicted in Figure 4 because the β 5-6 loop was not modelling in the OC43 crystal structure. Indeed, many of the HE structures obtained by X-ray crystallography have disordered regions. We have added an approximation of this missing loop to figure 4D in order to better highlight the structural difference to HKU1, where this loop is genuinely absent. The corresponding figure legend has also been updated.

In reporting the structure determination, a measure of agreement between the map and model should be provided such as the map-model FSC because good geometry as reported can be achieved without reference to the map. I note that an EMRinger score is provided.

We apologize for this unintentional omission. Table S1 has now been updated with the model resolution (0.5 FSC) obtained from the final phenix real-space refinement job.

The authors should confirm that their starting model for 3D classification was low-pass filtered to 40 Å.

A 40 Å resolution starting model was generated using the 'molmap' command in UCSF Chimera (lines 308-310). To be doubly certain that no model bias was introduced, the starting model was also low-pass filtered to 40 Å resolution at the start of the Relion 3D classification job.

Reviewer #4 (Remarks to the Author):

The manuscript from Hurdiss et al reports the 3.4 Å cryo-EM structure and analysis of the HKU1-CoV haemagglutinin esterase (HE) ectodomain. The structure reveals that loops in the lectin domain have been truncated and are no longer capable of binding to glycans, possibly as a result of adaptation to replication in humans. The esterase domain (ED) is intact and is similar to other coronavirus EDs. The authors also perform a mass spec analysis of the N-linked glycosylation, which provides quantitative information of the glycan shield. It appears that the HKU1 HE lectin domain has acquired additional glycosylation sites in the truncated loops, perhaps to shield this region from recognition by the humoral immune system.

The manuscript is clearly written and the figures are excellent. Obtaining a 3.4Å cryo-EM reconstruction on a relatively small (~80 kDa) protein is commendable, and the map and model stats are very good.

We thank the reviewer for their positive assessment of our manuscript.

A weakness is that the structure did not provide much new information that could not have been obtained from a homology model derived from the HKU1 HE sequence and related HE structures from MHV and BCoV, but the structural analysis and the insights into evolution of the HE protein were interesting.

Interestingly, there was disparity between the positioning of the lectin domain loops in the homology model and experimental density, reaching a maximum distance of 6.5 Å for the β 4- β 5 loop and 7.5 Å for the β 7- β 8 loop, both of which are N-glycosylated. The improved accuracy provided by our experimentally derived model now puts us in a strong position to start investigating the structural basis for HE targeting neutralising antibodies, particularly those which target the lectin domain.